# Effects of Traveling Magnetic Field on Interfacial Microstructure and Mechanical Properties of Al/Mg Bimetals Prepared by Compound Casting

**DOI:** 10.3390/ma18174077

**Published:** 2025-08-31

**Authors:** Qiantong Zeng, Guangyu Li, Jiaze Hu, Wenming Jiang, Xiuru Fan, Yuejia Wang, Xiaoqiong Wang, Xing Kang

**Affiliations:** 1School of Materials Science and Engineering, Dalian University of Technology, Dalian 116024, China; zqt_185@163.com (Q.Z.); dgwangyuejia@mail.dlut.edu.cn (Y.W.); xqw@mail.dlut.edu.cn (X.W.); xingkangx@163.com (X.K.); 2Ningbo Institute of Dalian University of Technology, Ningbo 315000, China; 3School of Mechanical Engineering and Automation, Dalian Polytechnic University, Dalian 116034, China; dpuhjz@163.com (J.H.); fanxiuru@dlpu.edu.cn (X.F.); 4State Key Laboratory of Materials Processing and Die & Mould Technology, School of Materials Science and Engineering, Huazhong University of Science and Technology, Wuhan 430074, China

**Keywords:** Al/Mg bimetals, traveling magnetic field, compound casting, interfacial microstructure, mechanical properties

## Abstract

In this work, the Al/Mg bimetals were prepared by traveling magnetic field (TMF)-assisted compound casting, and the effects of current intensity on the interfacial microstructure and mechanical properties of the Al/Mg bimetals were investigated. The results revealed that the Al/Mg bimetallic interface without the TMF consisted of an Al-Mg intermetallic compounds (IMCs) area (Al_3_Mg_2_ + Al_12_Mg_17_ + Mg_2_Si particles) and Al-Mg eutectic area (Al_12_Mg_17_ + δ-Mg). There was no change in the interfacial phase compositions with the TMF, but the interface thickness initially decreased and then increased with the increase in the TMF current, and the distribution of Mg_2_Si became more uniform, dendrites become smaller, and dendritic arms fragment. The shear strength improves from 17 MPa without the TMF to 27 MPa with the TMFed-60 A, which was increased by 58.8%. This enhancement occurs because cracks are deflected by uniformly distributed Mg_2_Si particles and do not coalesce into main cracks, ultimately fracturing in the eutectic region, which increases the length of the crack propagation path and thereby improves the shear strength of the Al/Mg bimetals.

## 1. Introduction

Al/Mg bimetallic composite materials combine the respective performance advantages of aluminum and magnesium alloys [1,2,3]. These composite materials can significantly reduce overall weight while maintaining structural strength and functionality through effective bonding of the two metals, thereby achieving lightweight manufacturing objectives. Such materials exhibit tremendous potential in aerospace, rail transportation, automobiles, the Low-Altitude Economy, and electronic product casings [4]. Current methods for preparing Al/Mg bimetals primarily include rolling [5], welding [6], and compound casting [7].

Compound casting offers several advantages, including the ability to fabricate complex-shaped and large-sized bimetallic components, ease of achieving metallurgical bonding, potential for heat treatment strengthening, and low cost, giving it broad application prospects [8,9,10]. However, several challenges arise when employing compound casting processes to prepare Al/Mg bimetals, with the bimetallic interface microstructure exhibiting problems such as coarse grains, non-uniform composition distribution, and excessive interface layer thickness [11]. Furthermore, cracking readily occurs with external forces since the bonding interface consists of brittle Al-Mg IMCs [12].

Researchers have conducted extensive studies to address these challenges, with the main research directions including adding intermediate layers [13,14,15,16,17,18], alloying [19,20,21,22], oxide film removal [23], secondary rolling [24], insert thermal modification [25], and applying external fields [26,27,28,29,30]. Compared with other methods, external fields offer unique advantages. Unlike adding intermediate layers, external fields do not require changing interface composition while enhancing interface strength. Compared to secondary rolling, external fields are applied during liquid solidification processes, resulting in more significant control effects. In contrast to expensive rare earth alloying methods, applying external fields can reduce the cost of interface control. In addition, external fields can simultaneously improve both matrix and interface microstructural properties. Currently, external fields employed in bimetals mainly include ultrasonic, vibration, and electromagnetic fields.

Electromagnetic stirring technology induces electromotive force and induced currents within the melt by applying alternating magnetic fields externally to the melt. Lorentz forces are generated through the interaction between these two factors, driving metal melt flow and improving solidification conditions, refining grains, and reducing casting defects [31,32,33]. Various electromagnetic fields such as low-frequency electromagnetic fields, pulsed magnetic fields, and composite magnetic fields have been widely applied in the preparation of various single alloys [34,35,36], but research on their application in Al/Mg bimetal interface control remains limited.

Huan et al. [37] investigated the effect of traveling and rotating coupled magnetic fields on an Al-7Si-Cu-Mg-Mn/steel bimetal interface. Melt flow induced by the magnetic field increased the Fe concentration gradient at the liquid–solid interface, promoting Fe diffusion from intermetallic phases to the liquid and accelerating Al_8_Fe_2_Si phase dissolution while retarding its growth. The interface layer thickness decreased from 5.1 ± 0.9 μm to 2.7 ± 0.2 μm, and shear strength increased from 41 ± 7 MPa to 56 ± 2 MPa. Yu et al. [38] studied the effect of spiral magnetic field on the interfacial microstructure and mechanical properties of Al/Mg bimetals. Mg_2_Si particles were refined and dispersed within the IMC region with electromagnetic forces. Oxide films became discontinuous with increased Mg_2_Si content in surrounding areas while remaining present, resulting in longer crack propagation paths and 29.1% improvement in shear strength. Yang Li et al. [39,40] effectively improved aluminum alloy wetting behavior on titanium substrates by applying a transverse alternating magnetic field during laser fusion brazing of AA5083/TA5. The original heterogeneous microstructure transformed into uniformly dispersed fine equiaxed grains. Additionally, interface defects were reduced, interface grain growth orientation became more diverse, interface hardness distribution became more uniform, and the average tensile strength and elongation of joints improved from 230.3 MPa and 2.5% to 277.1 MPa and 4.3%, representing increases of 20.3% and 72.0%, respectively. However, systematic research on the effects of traveling magnetic field (TMF) and their parameters on the interfacial microstructure and properties of sand-cast Al/Mg bimetals remains lacking.

In this study, TMF-assisted compound casting was used to prepare Al/Mg bimetals, and the effects of TMF’s current intensity on the interfacial microstructure and properties were investigated.

## 2. Materials and Methods

### 2.1. Material Preparation

In the experiments, A356 aluminum alloy and AZ91D magnesium alloy were used to manufacture Al/Mg bimetals, and their detailed compositions are shown in Table 1. A356 aluminum alloy served as the solid insert material. The inserts with a diameter of 15 mm and height of 110 mm were obtained from A356 aluminum ingots through wire cutting then sequentially polished with 220, 400, 1000, and 2000 grit silicon carbide sandpaper. Finally, the A356 inserts were cleaned with anhydrous ethanol in an ultrasonic cleaner for 10 min and preheated to 100 °C in a drying oven. AZ91D magnesium alloy was melted in a resistance furnace at 750 °C. In this work, the AZ91D pouring temperature, sand mold preheating temperature, and insert preheating temperature were 730 °C, 100 °C, and 100 °C, respectively.

Figure 1a shows a schematic diagram of the TMF-assisted compound casting system, which mainly consists of a TMF generator device and sand mold. The TMF was immediately activated after pouring completion. Four types of samples were prepared: Al/Mg bimetallic samples without the TMF and three additional samples treated at current intensities of 30 A, 60 A, and 90 A and correspondingly designated as TMFed-30 A, TMFed-60 A, and TMFed-90 A.

### 2.2. Characterization

The red box in Figure 1c shows the sample cutting position for microstructural characterization. Samples were polished with 1.5 μm diamond paste and etched with 4% nitric acid alcohol solution for 5 s after removing machining marks by sandpaper grinding. The microstructure and chemical composition of Al/Mg bimetals were analyzed using a JSM-IT800 scanning electron microscope (SEM, JSM-IT800, JEOL, Tokyo, Japan) equipped with energy dispersive X-ray spectroscopy (EDS). Microhardness distribution across the Al/Mg bimetallic interface was tested using a Vickers hardness tester (250HV-5BZ, Hongjian, Ningbo, China) with a loading time of 10 s and a load of 300 g. The hardness measurements were conducted starting from the Al substrate to Mg substrate and proceeding along the direction perpendicular to the interface, with a test step of approximately 150 μm. The shear strength of Al/Mg bimetals was measured using a universal testing machine (Instron 100 kN, Instron, Norwood, MA, USA) with a loading rate of 0.5 mm/min, and the schematic diagram of shear strength measurement is exhibited in Figure 1b. For each condition, three specimens were tested, and the average value was taken as the shear strength of the sample. Shear performance can be calculated according to Equation (1):(1)$$S=\frac{F}{\pi dh}$$
where *F* is the maximum force loaded during compression, *d* is the diameter of A356 insert, and *h* is the height of the tested specimen. In this work, the diameter of A356 insert was 15 mm, and the specimen height was 3 mm. Additionally, fracture surfaces and crack morphologies after shear strength testing, as well as interface indentations, were observed through SEM.

## 3. Results

### 3.1. Microstructure Evolution

Figure 2 and Figure 3 present the microstructure of the bimetallic interface under different TMF conditions. As observed in Figure 2a, an interface was formed between the aluminum matrix and magnesium matrix. According to the area scanning results (Figure 2(a1–a3)), the interface is non-uniform, with higher concentrations of Al and Si elements near the aluminum side, while the magnesium side exhibits elevated Mg content. According to the variation in interface thickness (as shown in Figure 4), the thickness initially decreases and then increases with the application of TMF and the rise in current intensity, with specific values of 2267, 1960, 2379, and 2437 μm, respectively. However, the eutectic layer thickness continues to increase, as shown in Figure 4. Furthermore, it can be observed that with the increase in current intensity, dendrites become progressively smaller, dendritic arms gradually fragment, and the interface transforms from flat to irregular.

For detailed observation of the morphology of various reaction layers and determination of the phase composition of interfacial layers, different regions were magnified, and the results are presented in Figure 5. The phase composition of the interface was determined by EDS point analysis and the Al-Mg and Mg-Si phase diagrams, as shown in Table 2. According to the results, the Al/Mg interface can be roughly divided into two regions: the eutectic layer (Al_12_Mg_17_ + δ-Mg eutectic structure) and compound layer (Al_12_Mg_17_ + Mg_2_Si and Al_3_Mg_2_ + Mg_2_Si), collectively referred to as the intermetallic compound layer), which is consistent with the research results of others [41,42].

In the Al_3_Mg_2_ layer of specimens without the TMF, Mg_2_Si particles exhibit a vermicular and network-like distribution (Figure 5c). The Al_12_Mg_17_ layer contains abundant Mg_2_Si with a network-like distribution (Figure 2a), but some uniform Mg_2_Si is also present (Figure 5b). The eutectic region consists of an Al_12_Mg_17_ + δ-Mg eutectic structure, with almost no Mg_2_Si particles observed, while densely distributed Mg_2_Si particles are present at the boundaries between the eutectic layer and IMCs layer, as shown in Figure 2a.

The TMF-treated interface retained the same phase composition, but its phase distribution and size underwent significant changes, as shown in Figure 5 and Figure 6. After the application of TMF, the average size of Mg_2_Si particles in the Al_3_Mg_2_ layer exhibited a slight increase, and their volume fraction increased. Moreover, as shown in Figure 5c,f–j, the distribution of Mg_2_Si within the Al_3_Mg_2_ layer became more uniform under TMF conditions. In contrast, the Mg_2_Si particles in the Al_12_Mg_17_ layer were refined and showed an increased volume fraction compared with the condition without the TMF, indicating that Mg_2_Si was more densely dispersed across the Al_12_Mg_17_ matrix, with the most pronounced effect observed at 60 A, as shown in Figure 3a and Figure 5h,i. In addition, the eutectic region showed no significant changes under all four conditions. These results indicate that TMF with appropriate current is beneficial for the refinement of Mg_2_Si in the Al_12_Mg_17_ layer and the distribution of Mg_2_Si throughout the entire intermetallic compound layer.

### 3.2. Mechanical Properties

#### 3.2.1. Microhardness

The Vickers microhardness distribution and Vickers indentations of the Al/Mg interface are shown in Figure 7. The microhardness of the Al/Mg interface was significantly higher than that of the A356 and AZ91D matrices. The microhardness values of Al-Mg IMCs without the TMF ranged between 240 HV and 271 HV, while the microhardness of the eutectic ranged between 184 HV and 196 HV, significantly lower than that of Al-Mg IMCs. A notable feature in the hardness distribution is that the Al-Mg IMCs distribution at 90A is extremely non-uniform, with a minimum value of 214 HV and a maximum value of 285 HV, which is consistent with the microstructural observations in Figure 3b: some regions exhibit uniform distribution, while others show non-uniform distribution. Compared to specimens without the TMF, the maximum hardness increased, while the minimum hardness decreased. The hardness improvement may be attributed to Mg_2_Si segregation.

The SEM images in Figure 8 show typical Vickers indentation micrographs of the Al_3_Mg_2_ layer at the Al/Mg interface. Compared to specimens without TMF treatment, TMF-treated specimens showed reduced crack lengths at indentation edges. As the current increases, the number of cracks gradually decreases, with the fewest cracks observed at 90A because the indentation occurred in a region with uniform Mg_2_Si distribution at 90A. Additionally, crack propagation paths were observed to be deflected by refined Mg_2_Si particles, as indicated by red arrows near Mg_2_Si in Figure 8, demonstrating that dispersed and refined Mg_2_Si particles can suppress crack propagation during deformation, thereby improving the bonding strength of Al/Mg bimetals [43].

#### 3.2.2. Shear Strength

Figure 9a,b show the shear strength test results of Al/Mg bimetals. Specimens without the TMF exhibited an average shear strength of 17.0 MPa. After 30A, 60A, and 90A TMF treatments, specimens showed shear strengths of 21.1 MPa, 27.0 MPa, and 22.3 MPa, respectively, representing improvements of 24.1%, 58.8%, and 31.1% compared to untreated specimens. Furthermore, following the same measurement scheme used for the bimetal, the shear performance of A356 and AZ91D was also evaluated. It can be observed that, compared to the Al/Mg bimetal, the individual base metals exhibit greater strain and higher shear strength during the shear process. This can largely be attributed to the presence of brittle Al-Mg intermetallic compounds at the interface [26].

The fracture morphology after shear strength measurement observed by SEM is shown in Figure 10. Distinct cleavage planes and river patterns were observed on both fracture surfaces with and without TMF treatment. The specimens without the TMF exhibited extensive flat cleavage planes and river patterns, whereas the fracture surface of TMFed-30A displayed more tear ridges with increasingly irregular cleavage planes, as illustrated in Figure 10(b,b1,c) shows the fracture surface at 60A, which presents a distinct layered structure, indicating that more crack branches were generated during propagation, thereby increasing the length of the crack propagation path. As for TMFed-90 A, the upper portion demonstrates large and flat cleavage planes, while the lower portion shows a certain number of ridges (Figure 10(d1)). These results demonstrate that both specimens with and without the TMF failed due to typical brittle fracture modes; however, with the introduction of TMF, the length of the crack propagation path increased.

Figure 11 shows the crack propagation paths of TMFed-0 A, TMFed-30 A, TMFed-60 A, and TMFed-90 A specimens after shear strength measurement. It can be observed that regardless of whether TMF is applied, cracks initiate from the IMC region near the A356 matrix. Based on the crack propagation direction, they can be roughly classified into two types: the first type of cracks propagate almost vertically to the bottom after initiation, designated as vertical cracks, while the second type propagate laterally from the aluminum matrix side toward the magnesium side, designated as lateral cracks.

For the first type of vertical cracks, there are virtually no changes whether TMF is applied or not; however, the lateral propagating cracks behave differently. When TMF is not applied, multiple initial lateral propagating cracks coalesce during propagation through the IMC region, forming coarser main cracks as shown in Figure 11a. Subsequently, the main cracks rapidly propagate through the brittle Al_12_Mg_17_ region, leading to fracture. In contrast, specimens with the TMF show that the total path of lateral cracks becomes longer, extending to more distant eutectic regions as illustrated in Figure 11b–d. Moreover, during propagation, the cracks mainly exist in finer forms with minimal coalescence, and crack inhibition and deflection are observed when passing near Mg_2_Si particles, as shown in Figure 11(b1,c1,d1).

## 4. Discussion

### 4.1. The Effect of TMF on the Microstructure of Al/Mg Bimetallic Interface

It can be observed that TMF induces numerous changes at the interface, and the underlying reasons are explored below. First, the Al/Mg interface after the TMF exhibits an irregular morphology, as shown in Figure 2b and Figure 3a,b. This irregularity can be attributed to the forced convection of the AZ91D melt induced by the TMF, which enhances melt fluidity and continuously erodes the A356 insert surface layer, consequently creating an irregular Al/Mg interface with more Al and Si elements entering the interface. Additionally, the eutectic layer thickness continuously increases with current intensity, as illustrated in Figure 4. Given that the eutectic layer consists of (Al_12_Mg_17_ + δ-Mg) and is located near the Mg matrix side, this phenomenon is likely ascribed to TMF increasing the Al content at the interface and enabling Al elements to diffuse farther, where they react with Mg in the liquid AZ91D to produce more eutectic structure.

Furthermore, under TMF treatment conditions, the distribution of Mg_2_Si in the IMC region becomes more uniform, accompanied by the refinement of Mg_2_Si particles in the Al_12_Mg_17_ layer. According to the electromagnetic induction law, relative motion exists between the varying magnetic field and the melt, generating induced currents within the melt; the interaction between these induced currents and the alternating magnetic field subjects the melt to electromagnetic forces [44,45], thereby producing forced convection. During interface formation, after the acicular eutectic Si in A356 melts, the forced convection causes Si atoms to diffuse farther and more uniformly, ultimately dispersing throughout the IMC region where they combine with Mg atoms to form dispersed Mg_2_Si, as shown in Figure 12. As for the growth of Mg_2_Si in the Al_3_Mg_2_ layer, as mentioned above, the forced convection induced by the TMF increased the concentrations of Al and Si elements at the interface. According to the study on the formation mechanism of the Al/Mg interface [10], the Al_3_Mg_2_ layer is the first region to solidify at the interface, which means that it may experience a shorter time to TMF compared with the Al_12_Mg_17_ layer. As a result, the dispersion of Si in the Al_3_Mg_2_ layer is less sufficient than in the Al_12_Mg_17_ layer, while the Si content is higher than that without the TMF treatment. Consequently, Mg_2_Si exhibits slight growth.

### 4.2. Strengthening Mechanism of the Al/Mg Bimetal

According to the hardness distribution curves, the Vickers hardness of the Al matrix is approximately 60 HV, while that of the Al_3_Mg_2_ layer is around 250 HV, indicating a severe mismatch in their plastic deformation capabilities. Since the hardness distribution patterns remain similar regardless of whether TMF is applied, initial cracks invariably form at the boundary between the IMCs and A356 matrix during the shearing process [43].

Without the TMF, lateral cracks aggregate into main cracks during propagation under the influence of segregated Mg_2_Si particles. This occurs because the thermal expansion coefficients of Al_3_Mg_2_, Al_12_Mg_17_, and Mg_2_Si differ significantly, leading to stress concentration in regions where Mg_2_Si particles accumulate [46]. When cracks encounter these locations, they readily coalesce to form main cracks that propagate rapidly, resulting in cleavage fracture as shown in Figure 11a, which consequently reduces the shear strength of the Al/Mg bimetals, as shown in detail in Figure 13a. After TMF application, the distribution of Mg_2_Si in the compound region is improved. Although the average size of Mg_2_Si particles in the Al_3_Mg_2_ layer slightly increased after the application of TMF, this effect can be compensated by the enhanced dispersion strengthening resulting from the improved uniformity and increased number of particles. During lateral crack propagation, dispersed Mg_2_Si particles deflect and split the cracks into branches, successfully penetrating through the IMC area and ultimately fracturing in the eutectic region, as illustrated in Figure 13b. This extends the crack propagation path, and longer cracks can consume more energy during the deformation process, which is beneficial for improving the shear strength of the Al/Mg bimetals [47]. The positive effect of magnetic field–induced forced convection on element transport exists only within a certain range. Within this range, the elements can diffuse more uniformly; however, beyond it, intense convection may lead to element segregation again. At 90 A, the strong forced convection caused by the high current intensity led to a mixed interfacial microstructure, exhibiting a mixture of homogeneous and heterogeneous regions. As shown in Figure 3b, the Al_3_Mg_2_ region contained many large white areas without Mg_2_Si distribution. while the Al_12_Mg_17_ region exhibits a uniform distribution of Mg_2_Si, as illustrated in Figure 5k. This is similar to the extremely non-uniform hardness distribution at 90A, shown in Figure 7. This morphology is intermediate between the 0 A and 60 A conditions. Therefore, its shear performance also falls between these two conditions, showing a decrease compared to the 60A condition.

## 5. Conclusions

This study employed traveling magnetic field assistance to prepare Al/Mg bimetals and systematically investigated the effects of traveling magnetic fields at different current intensities on interface microstructural evolution and mechanical properties. The main conclusions are as follows:Al/Mg bimetallic interfaces without the TMF showed non-uniform microstructures composed of Al-Mg IMC regions (Al_3_Mg_2_ + Mg_2_Si, Al_12_Mg_17_ + aggregated Mg_2_Si) and Al-Mg eutectic areas (Al_12_Mg_17_ + δ-Mg). TMF increased Al/Mg interface eutectic layer thickness, increasing with current intensity, and interface morphology changed from flat to irregular.The traveling magnetic field promotes the refinement and dispersion of Mg_2_Si particles. An appropriate current intensity (such as 60 A) can effectively refine the Mg_2_Si particles in the Al_12_Mg_17_ layer and achieve a relatively uniform distribution without altering the phase composition in the IMC region, thereby suppressing Mg_2_Si segregation and improving interfacial homogeneity.TMF improved Al/Mg interface shear strength. TMFed-60 A’s Al/Mg bimetals achieved shear strength of 27 MPa, representing a 58.8% improvement over Al/Mg bimetals without the TMF. This can be attributed to more uniform microstructures, and uniformly dispersed Mg_2_Si particles, which contribute to longer crack propagation paths.The TMF-assisted compound casting demonstrates significant potential for industrial applications. The technique achieved a remarkable 58.8% improvement in shear strength with the TMFed-60 A treatment while optimizing microstructures without changing phase compositions. Moreover, the process can be easily integrated into existing casting processes with minimal additional equipment, thus providing a cost-effective solution for enhancing bimetals in engineering applications.

## Figures and Tables

**Figure 1 materials-18-04077-f001:**
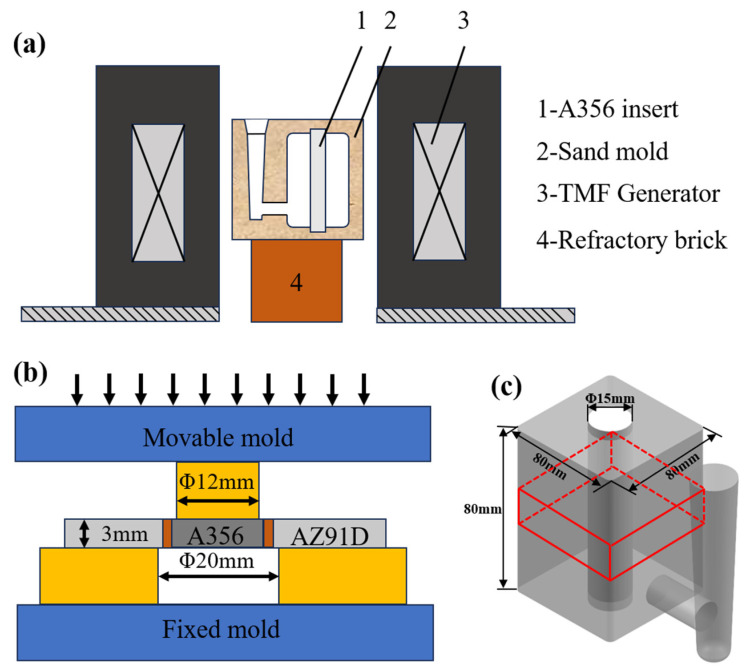
Schematic diagrams: (**a**) the TMF assisted experimental system; (**b**) schematic diagram of the shear test; (**c**) sampling location for microstructural analysis (marked in red).

**Figure 2 materials-18-04077-f002:**
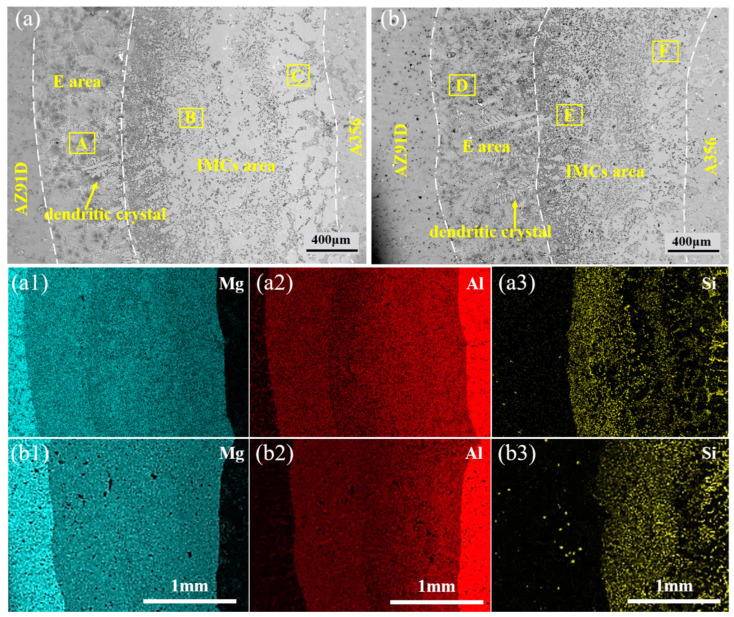
Typical morphologies and EDS maps of the Al/Mg interfaces: (**a**,**a1**–**a3**) without the TMF; (**b**,**b1**–**b3**) TMFed-30 A.

**Figure 3 materials-18-04077-f003:**
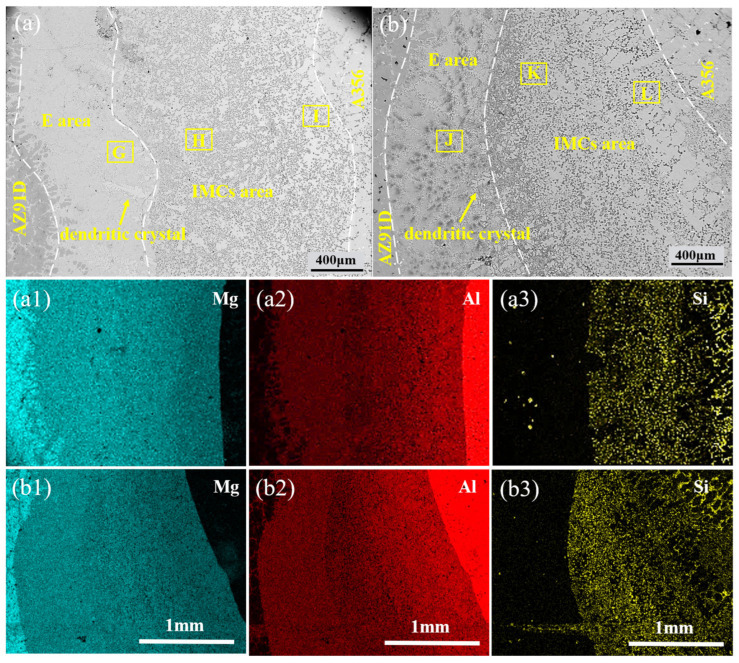
Typical morphologies and EDS maps of the Al/Mg interfaces: (**a**,**a1**–**a3**) TMFed-60 A; (**b**,**b1**–**b3**) TMFed-90 A.

**Figure 4 materials-18-04077-f004:**
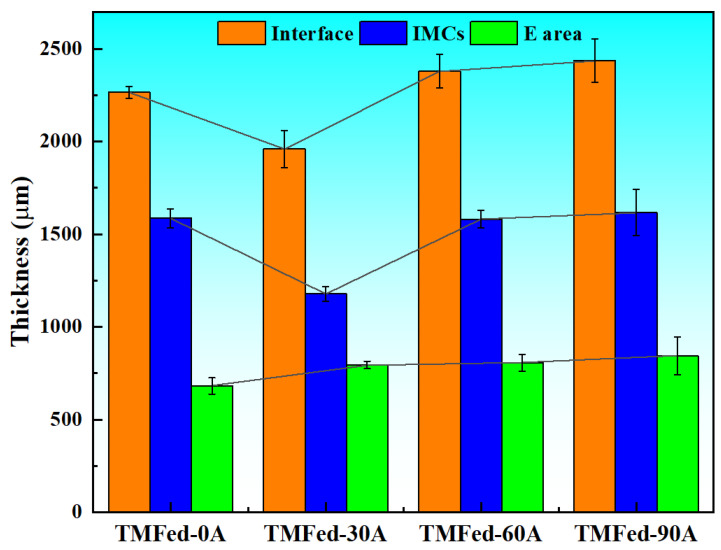
Proportion of different reaction layers.

**Figure 5 materials-18-04077-f005:**
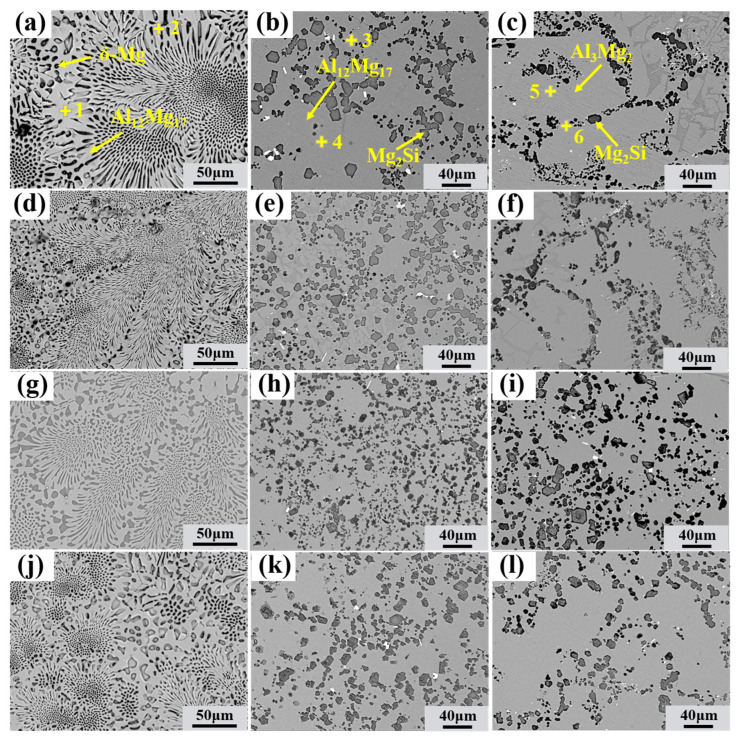
SEM images of the Al/Mg bimetal interface: (**a**–**c**) corresponding to the marked region A–C in Figure 2a; (**d**–**f**) corresponding to the marked region D–F in Figure 2b; (**g**–**i**) corresponding to the marked region G–I in Figure 3a and (**j**–**l**) corresponding to the marked region J–L in Figure 3b.

**Figure 6 materials-18-04077-f006:**
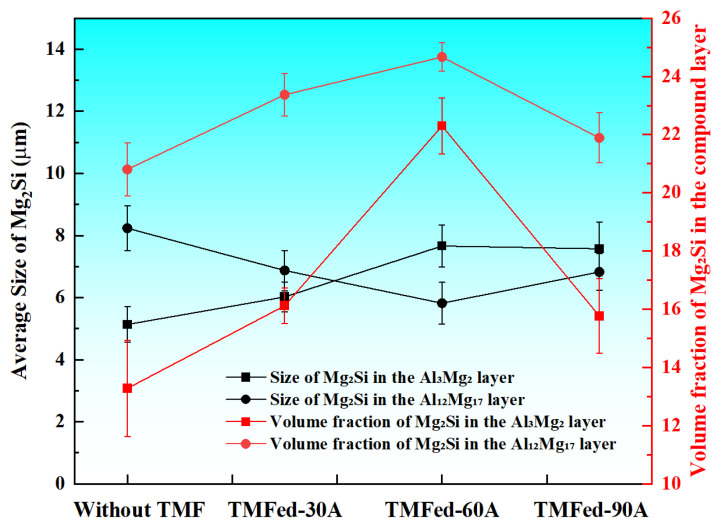
Quantitative analysis of the Mg_2_Si phase in the IMC area of the interface.

**Figure 7 materials-18-04077-f007:**
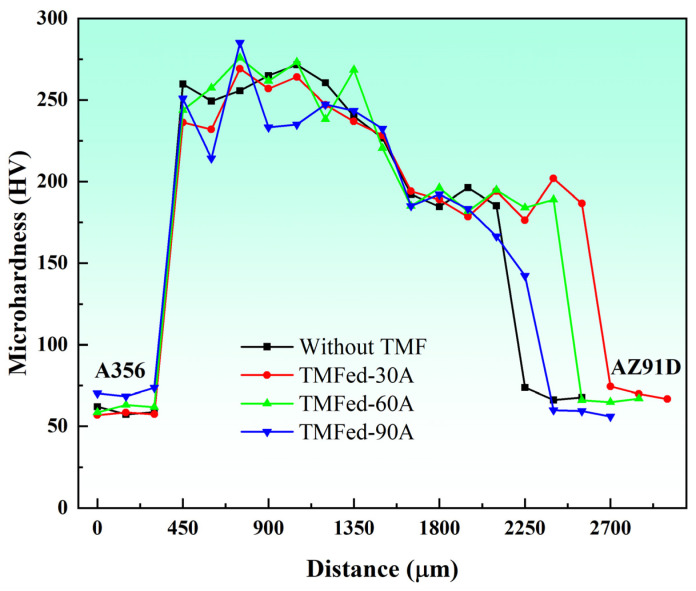
Vickers hardness of the Al/Mg bimetals.

**Figure 8 materials-18-04077-f008:**
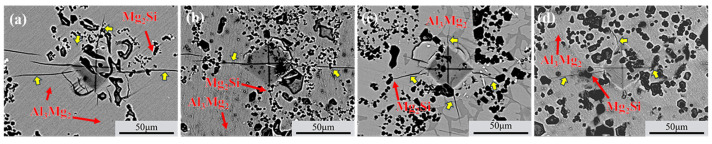
SEM image illustrates the cracks and Mg_2_Si particles at the edges of Vickers indentation at Al_3_Mg_2_ area: (**a**) without TMF; (**b**) TMFed-30 A; (**c**) TMFed-60 A; (**d**) TMFed-90 A.

**Figure 9 materials-18-04077-f009:**
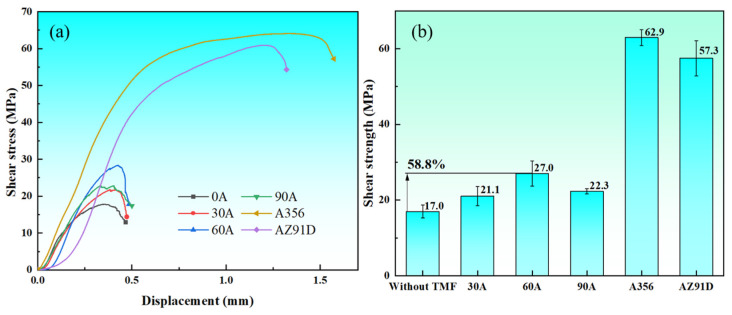
Results of shear strength testing: (**a**) stress-displacement curves; (**b**) average shear strength.

**Figure 10 materials-18-04077-f010:**
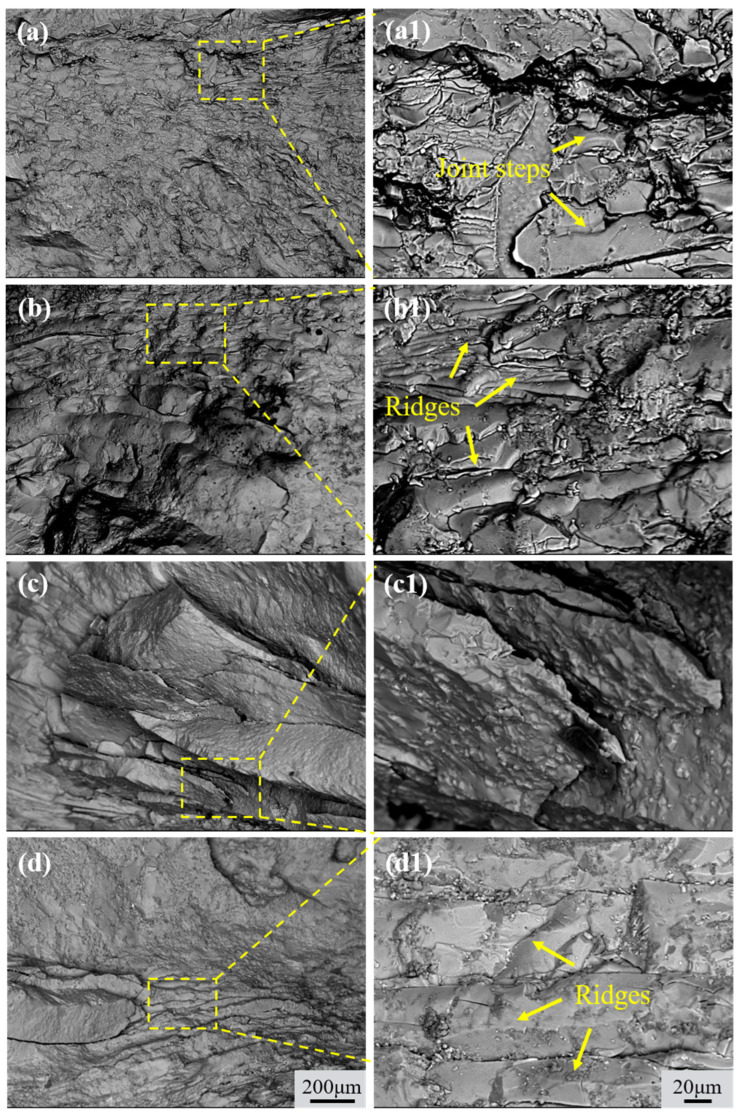
SEM morphologies of the fracture surfaces of the Al/Mg bimetals at Al substrate side: (**a**,**a1**) without the TMF; (**b**,**b1**) TMFed-30 A; (**c**,**c1**) TMFed-60 A; (**d**,**d1**) TMFed-90 A.

**Figure 11 materials-18-04077-f011:**
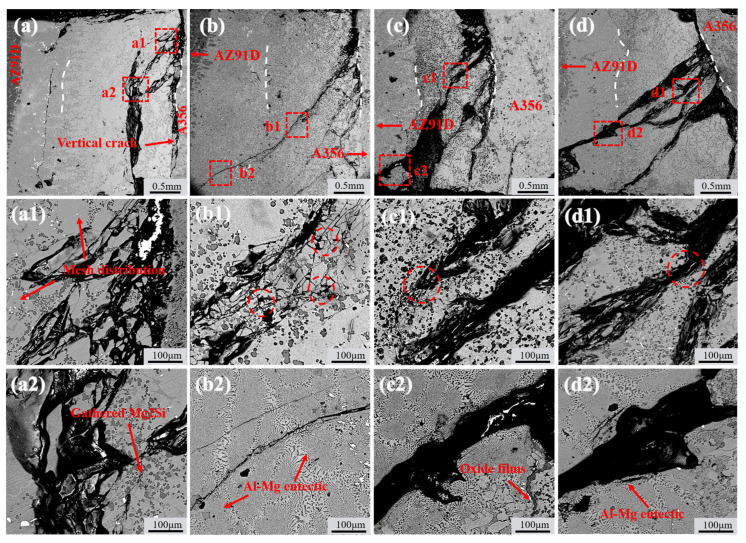
Crack propagation paths at the Al/Mg interfaces after shear strength measurements: (**a**,**a1**,**a2**) without the TMF; (**b**,**b1**,**b2**) TMFed-30 A; (**c**,**c1**,**c2**) TMFed-60 A; (**d**,**d1**,**d2**) TMFed-90 A.

**Figure 12 materials-18-04077-f012:**
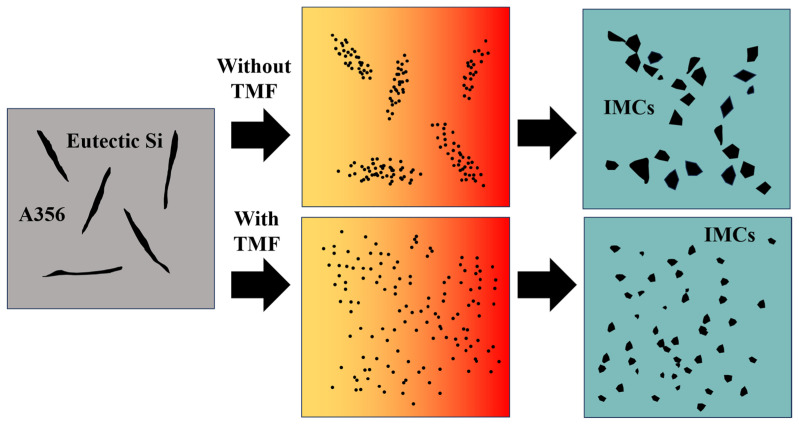
The influence mechanism of the TMF on the Mg2Si phase.

**Figure 13 materials-18-04077-f013:**
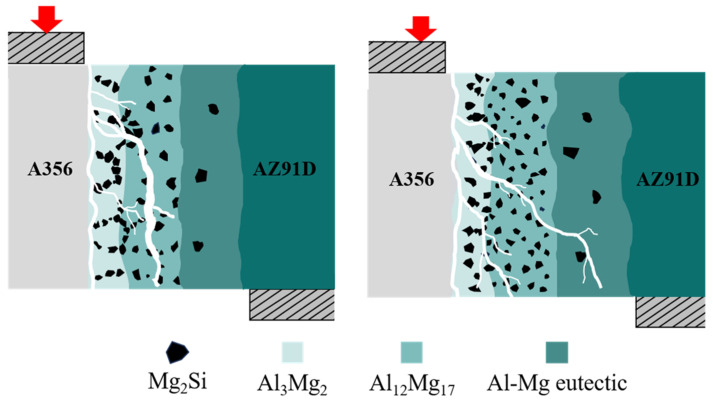
Fracture mechanism of the Al/Mg bimetals: (**a**) without the TMF, (**b**) with the TMF. (The red arrows indicate load).

**Table 1 materials-18-04077-t001:** The compositions of A356 and AZ91D.

Alloy			Mass	Fraction (%)			
	Mg	Al	Si	Ti	Mn	Fe	Zn
A356	0.439	Bal.	6.81	0.017	-	00.205	-
AZ91D	Bal.	9.08	-	-	0.62	-	0.23

**Table 2 materials-18-04077-t002:** EDS results of different locations in Figure 5a–c.

Point No.	Element Compositions (at.%)			Possible Phase
	Mg	Al	Si	
1	61.92	38.08	-	Al_12_Mg_17_
2	90.71	9.29	-	δ-Mg
3	55.39	44.61	-	Al_12_Mg_17_
4	54.87	-	45.13	Mg_2_Si
5	38.37	61.63	-	Al_3_Mg_2_
6	66.48	-	33.52	Mg_2_Si

## Data Availability

The original contributions presented in this study are included in the article. Further inquiries can be directed to the corresponding authors.

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
