# Peer review of "Effects of Traveling Magnetic Field on Interfacial Microstructure and Mechanical Properties of Al/Mg Bimetals Prepared by Compound Casting"

_materials, 2025, doi:10.3390/ma18174077_

Round 1
Reviewer 1 Report
Comments and Suggestions for Authors
The microstructure and shear strength of the A356/AZ291D bimetal were improved by applying a traveling magnetic field during casting. Increasing the density current to 60 A resulted in refinement and improved distribution of the Mg2Si particles, and in consequence, the shear strength increased 58.8% and the brittleness of the interface was reduced.
The manuscript presents good quality and interesting findings, but there are a few points that require attention:
a) There are typos throughout the manuscript;
b) Figures 2 and 3: Explain which element each color represents and improve contrast;
c) Provide the error/standard deviation of the thickness measures;
d) Measure the size and particle density of the Mg2Si particles in the compound layers. Quantitative results will strongly support the results;
e) Figure 10: Indicate where the A356 and AZ91D sides are;
f) Figure 12 was not mentioned in the text.
Reviewer 2 Report
Comments and Suggestions for Authors
Journal: Materials
Manuscript ID: materials-3826089
Type of manuscript: Article
Title: Effects of Traveling Magnetic Field on Interfacial Microstructure and
Mechanical Properties of Al/Mg Bimetals Prepared by Compound Casting
Authors: Qiantong Zeng, Guangyu Li *, Jiaze Hu, Wenming Jiang *, Xiuru Fan,
Yuejia Wang, Xiaoqiong Wang, Xing Kang
The authors investigate the influence of a magnetic field on the production of Al/Mg bimetals by casting. The method is elegant in its simplicity and therefore interesting. Research of bimetals is still very relevant today. To achieve the best strength characteristics, the authors vary the magnetic field current and investigate the structure and mechanical properties of the obtained samples. From the point of view of applicability, I would say that the strength of bimetals is very low. Nevertheless, the proposed method helped to increase strength by 58%. In addition, an interesting and justified mechanism of fracture based on crack deviation and branching is proposed. Overall, the work is of a high standard, and the comments are of a technical nature:
- Figures 2 and 3: the elements on the EDS maps should be specified.
- "Statistical analysis of the interfacial layer thickness reveals that with the application of TMF and increasing current intensity, it follows a pattern of initial decrease followed by increase, with thicknesses of 2267, 1960, 2379, and 2437 μm, respectively" - I would like to see the standard deviation. The same applies to Figure 4. Judging by the SEM images, the thickness of the layers varies quite significantly. Therefore, in order to draw conclusions about the real effect of the current, it is necessary to know the range of values.
- “Compared to the 30A, the Mg2Si particle distribution at the IMCs of TMFed-60A was more uniform, showing further overall refinement, although a certain number of larger Mg2Si particles appeared, as shown in Figures 5(h,i).” Unfortunately, I cannot agree with this. As I see it, in Figure 5f, the particle size is larger than in Figure 5c. And in Figure 5e, the size has not changed compared to Figure 5b, but the number has increased. Perhaps, in order to confirm your statement, it is necessary to provide the results of measurements of particle size and their volume fraction. Considering that these arguments are key to describing the fracture mechanism, this is a rather important point.
- The shear strength of the A356 and AZ91D alloys needs to be specified.
- The explanation of the fracture mechanism seems quite logical and reasonable. However, it is unclear why the strength decreased at 90 A.
Reviewer 3 Report
Comments and Suggestions for Authors
In this work, the effect of magnetic fields on the microstructure and mechanical properties of bimetals was studied. While the results are interesting, the authors should address the following issues before considering for publication.
1. In Table 1, please indicate whether the quantities are given in weight or in atomic percents .
2. Please provide additional detailos of the indentation tests.
3. "Statistical analysis of the interfacial layer thickness reveals..." What statistical analysis? Please elaborate.
4. "At 30A, the Mg2Si particle distribution became more uniform compared to 0A, with refined particle size,..."
Is it possible to give an estimation of the particle size? This could provide some quantitative insights.
5. How did the authors measure the microhardness?
6. How many specimens were tested for each case? I'm wondering whether the enhancements in shear stress is statistically consistent.
7. I suggest to add a brief ending paragraph to the Conclusions highlighting the relevance of using TMF in engineering applications.
Round 2
Reviewer 3 Report
Comments and Suggestions for Authors
The authors addressed all of my comments.